# First look by the Yutu-2 rover at the deep subsurface structure at the lunar farside

Jialong Lai 1,2, Yi Xu 1✉, Roberto Bugiolacchi 1,3, Xu Meng1,4, Long Xiao1,5, Minggang Xie 1,6, Bin Liu7, Kaichang Di7, Xiaoping Zhang1, Bin Zhou 8,9, Shaoxiang Shen8,9 & Luyuan Xu 1

The unequal distribution of volcanic products between the Earth-facing lunar side and the farside is the result of a complex thermal history. To help unravel the dichotomy, for the first time a lunar landing mission (Chang'e-4, CE-4) has targeted the Moon's farside landing on the floor of Von Kármán crater (VK) inside the South Pole-Aitken (SPA). We present the first deep subsurface stratigraphic structure based on data collected by the ground-penetrating radar (GPR) onboard the Yutu-2 rover during the initial nine months exploration phase. The radargram reveals several strata interfaces beneath the surveying path: buried ejecta is overlaid by at least four layers of distinct lava flows that probably occurred during the Imbrium Epoch, with thicknesses ranging from 12 m up to about 100 m, providing direct evidence of multiple lava-infilling events that occurred within the VK crater. The average loss tangent of mare basalts is estimated at 0.0040-0.0061.

[1] State Key Laboratory of Lunar and Planetary Sciences, Macau University of Science and Technology, Macau, China. [2] School of Science, Jiangxi University of Science and Technology, Ganzhou, China. [3] University College London, Earth Sciences, London, UK. [4] School of Civil Engineering, Guangzhou University, Guangzhou, China. [5] Planetary Science Institute, School of Earth Sciences, China University of Geosciences, Wuhan, China. [6] College of Science, Guilin University of Technology, Guilin, China. [7] State Key Laboratory of Remote Sensing Science, Aerospace Information Research Institute, Chinese Academy of Science, Beijing, China. [8] Key Laboratory of Electromagnetic Radiation and Detection Techonology, Chinese Academy of Sceience, Beijing, China. [9] Aerospace Information Research Institute, Chinese Academy of Science, Beijing, China. ✉email: yixu@must.edu.mo

Unraveling the shallow subsurface structure of the lunar mare offers the key to a better understanding of the local history of basaltic volcanism, an important process coupled to the Moon's thermal evolution[1]. The thickness and surface area of basalt layers can be used to constrain lava eruption volumes. A range of remote-sensing data including the study of impact craters morphology[2,3], the analysis of high-resolution gravity data[4], and the reflectance spectra of crater ejecta deposits[5–8] have contributed to developing the current model of lunar evolution.

Ground-penetrating radars on the lunar surface and radar sounders onboard orbiting spacecraft have helped to investigate the physical properties of the subsurface materials and their possible stratigraphy. The Apollo Lunar Sounder Experiment, part of the Apollo 17 mission, was the first instrument to detect deep subsurface reflectors corresponding to the interface between mare and bedrock[9] at average apparent depths of 1–1.6 km in Mare Serenitatis, Mare Crisium, and Oceanus Procellarum[9–11]. The apparent depth is defined as the propagation depth of a radar signal with the speed of light in the vacuum. Later, the Lunar Radar Sounder onboard the Kaguya spacecraft (SELENE) observed relatively shallow reflectors interpreted as subsurface boundaries between distinct basaltic rock layers in the nearside maria at apparent depths in the range of hundreds of meters[12–14].

Compared with the spaceborne radar experiment, the lunar-penetrating radar (LPR) onboard Chang'e-3 (CE-3) and Chang'e-4 (CE-4) rover have a much higher range resolution (1–2 m in the mare basalt for the 60 MHz channel), thus offering a unique opportunity to survey in greater detail the shallow subsurface of both the lunar nearside and farside[15–17]. CE-4 landed in the South Pole-Aitken (SPA) Basin, the largest known impact structure on the Moon and a key region ideally suited to address several outstanding geological questions as the impact might have even penetrated the entire lunar crust[18,19]. The radargrams produced from the data acquired by the CE-4 instrument reveal the basalt layer thickness of each lava eruption and the time sequence of surface modification events that occurred in the Von Kármán (VK) crater (Supplementary Fig. 1). In a broader context, this new information adds to our limited understanding of the igneous history of the SPA Basin, which is thought to have been significantly shorter and less extensive than its equivalent on the nearside[1,20]. The reason for the asymmetric distribution between the lunar sides is understood to relate either to differences in crustal thickness, to the abundance of radioactive elements, or to the geological consequences of the large SPA-forming impact itself[21–27].

In this work, we report the LPR results for the first 9 months derived from channel one (CH-1, 60 MHz) data and test our interpretations using LPR simulation. A stratigraphic model of the surveying area (landing coordinates based on Lunar Reconnaissance Orbiter terrain data: 177.5885°E, 45.4561°S, −5927 m[28,29]) was generated from the extracted reflectors profile, which suggests possible lava flows sources and a potentially complex buried topography. The local geological history of the CE-4 landing site is inferred based on the revealed stratigraphy.

## Results

**Geologic settings**. VK crater (171 km) lies within the SPA basin, an impact crater about 2500 km in diameter. The thermal history of the crater and its neighborhood thus should be interpreted within an atypical geological context[19–25]. During the Late Heavy Bombardment (LHB[30]) period, several giant impacts including Imbrium on the nearside and VK's northern neighbor, crater Leibnitz (245 km in diameter) were produced. Post LHB, the region underwent a relatively prolonged phase of lava infill, which

lasted about 200–600 Ma[21,31], with the youngest flows estimated between 3.15 and 3.6 Ga[19,21]. However, currently, no direct evidence of the volcanic history of VK crater indicates whether the mare deposits were formed by one episode of basaltic volcanism based on the uniform reflectance spectral characteristics or multiple lava-infilling events[19]. LPR can provide first-hand data to disclose the subsurface stratigraphy and constrain the thermal history.

VK's neighboring region is geologically highly complex: the map I-1047[31] and the inset[32] (Supplementary Fig. 1) show a superposition of impact morphologies spanning from the pre-Nectarian to the Copernican epochs. The neighboring impacts produced ejecta materials that punctuated the infill and post-infill phase of the VK crater. The time sequence of these craters is relevant to the interpretation of the stratigraphy at the CE-4 exploration path, which is analyzed in Supplementary Note 1.

**Lunar-penetrating radar results**. The penetrating depth of LPR CH-1 can reach up to ~330 m (Supplementary Note 2 and Supplementary Fig. 2), although the top section of the radar signals becomes saturated due to the strong coupling effects from the electromagnetic interaction with the metal in the rover. However, channel two (CH-2) of the LPR data, the center frequency of which is 500 MHz and can penetrate up to ~35 m[17], can be employed to complement the profile of the close-to-surface section[17] (Fig.1c). Here we focus on the LPR data analysis between 52 and 328 m.

The prominent and continuous subsurface reflectors A–E at depths of (A) 51.8 ± 1.1 m, (B) 63.2 ± 1.2 m, (C) 96.2 ± 3.2 m, (D) 130.2 ± 3.7 m, and (E) 225.8 ± 5.5 m can be observed both in the processed radar image and aggregated data traces displayed in terms of signal strength (dB, yellow line) (Fig. 1a). The horizontal reflectors appear relatively constant running parallel to the surface (see Fig. 1), except for the horizontal reflectors D that shows a gradual rise of 7.1 m in the right end. This is probably due to the change in subsurface topography, e.g., crater at depth of 130 m (see simulation results in Supplementary Fig. 4). From around waypoint 42, the reflector D becomes flat, because the rover conducted a local exploration mission to collect other scientific data at the end of the ninth month exploration with consequent little variation of the subsurface topography. Nonetheless, this localized and repeated sampling phase helps to constrain the consistency and reliability of the data gathering process.

The materials between the most prominent horizontals are rather uniform and strong radar echo are rare (e.g., A–B, B–C, C–D in Fig.1b); however, a couple of subtle features stand out at the bottom part region of the radargram using image enhancement technique. Some relatively short lines appear in the D–E strata and more continuous ones occur below reflector E, which are interpreted as ejecta at a different scale. As the thickness of stratum D–E is about 100 m, it is possible that it was formed by multiple lava eruption events interposed by small-scale ejecta deposits or thin regolith that was formed in the lull period: this geologically complex admix may reflect in the scattered features in the radar results as evident in Fig.1b. Alternatively, the large-scale ejecta layers at depths of over 200 m may produce relatively continuous signal discontinuities, but with more pronounced fluctuations than a well-defined interface.

**Simulation results**. To test our geological interpretation of the radar data, several subsurface models were designed for LPR simulation with various sets of loss tangent and permittivity values.

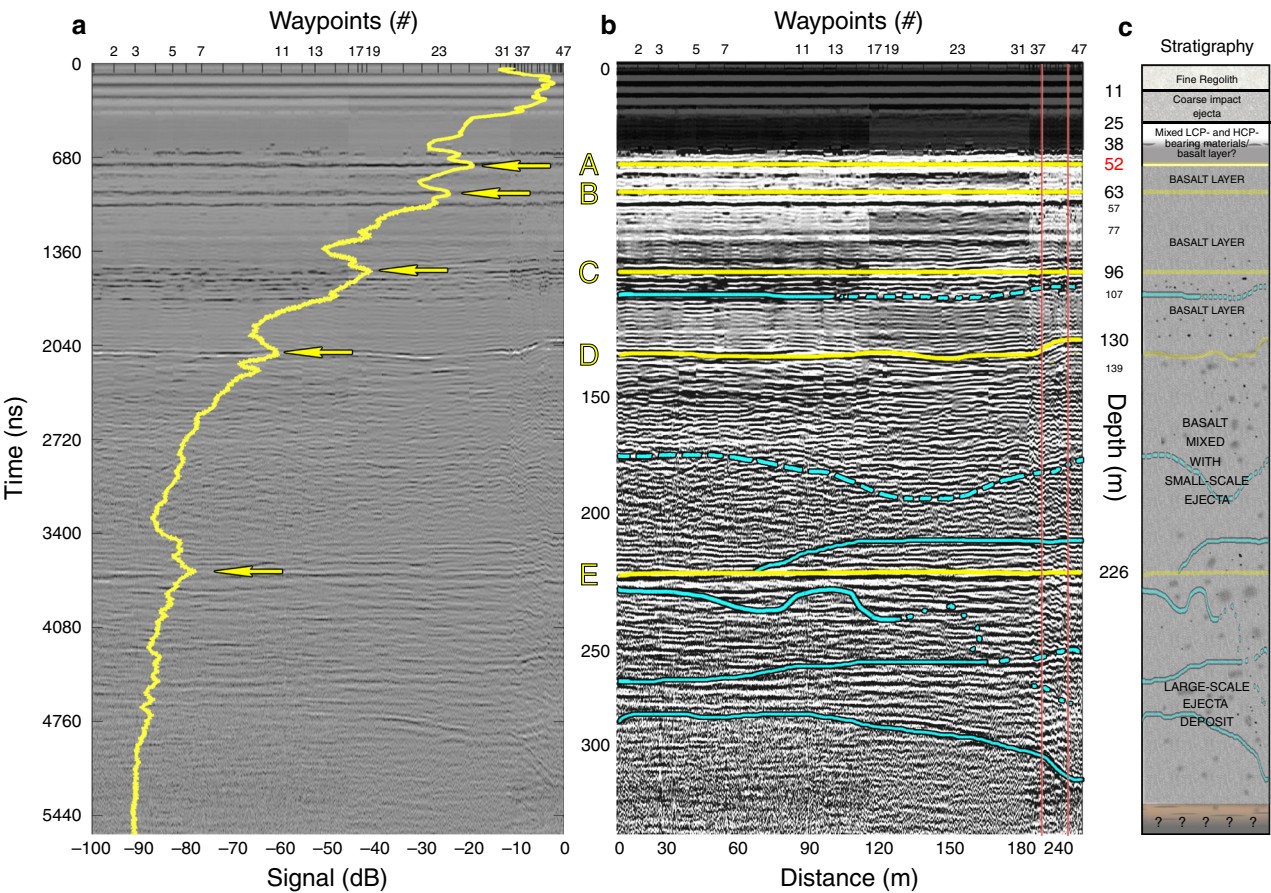

**Fig. 1 Chang'e-4 radargram for the 60 MHz channel. a** LPR CH-1 radargram of CE-4 landing site using an Automatic Gain Control (AGC) method[49] for amplitude compensation. The yellow line represents the aggregated data traces to show the enhanced subsurface echoes A–E. This approach also minimizes "anomalous" signal points along the travel path that might arise from the random distribution of rocks and debris among more heterogeneous layers. **b** Interpreted LPR CH-1 radargram of CE-4 landing site using image enhancement techniques. The cumulative length of the traversed path is 284.6 m. Yellow lines represent enhanced subsurface echoes A–E (**a**); light blue lines are subtle boundaries denoting differences in "stripe" directions and sharpness; dashed lines denote higher uncertainty in location. The two red vertical lines indicate waypoint 37 and 42, respectively. Please note that LPR data were not collected at a fixed speed. For example, the jump from 180 to 240 m at the end of X axis is because LPR CH-1 collected much fewer data at the end of the traverse path than at the beginning stage when rover traveled around a small crater. The enlarged images of the end of the traverse are given in Supplementary Fig. 3. **c** The interpreted stratigraphy structure inferred from the LPR results. $\varepsilon = 4.5 (\leq 52 \text{ m})$ and $6.5 (>52 \text{ m})$ is used for time-to-depth conversion.

The average loss tangent (the ratio of the imaginary and real part of permittivity, tanδ) in CE-4 site is inverted with three types of geometric spreading corrections, as shown in Fig. 2. For $R^2$ correction, $\tan\delta = 0.0060 \pm 0.0001$; for $R^3$ correction, $\tan\delta = 0.0051 \pm 0.0001$; for $R^4$ correction, $\tan\delta = 0.0041 \pm 0.0001$. In the case of rough interface ($R^3$ correction), we confirm the estimation value with the result inferred from the penetrating depth of LPR CH-1, which is 0.005 (see Supplementary Note 3). The first model (Fig. 3a) adopts the derived loss tangent value and models the subsurface structure underneath a short path, including regolith, ejecta from nearby craters, basalt layers admixed with small-scale ejecta, and large-scale ejecta formed in a time sequence from recent to remote. The simulation results show the clear boundary between the regolith and mare basalts (Fig. 3b). Also, both small- and large-scale ejecta could produce clear radar echoes, especially when larger debris is present at depths of over 200 m, which would produce flatter horizontal lines instead of large hyperbola shaped signals. The echoes of small-scale ejecta appear shorter and less continuous, comparable to the observations of the LPR results in D–E section. The simulation results show similar

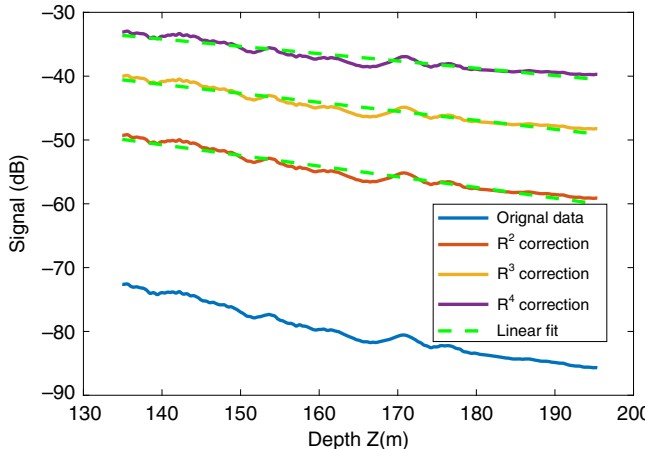

**Fig. 2 Depth vs. signal power profile.** $R^2$, $R^3$, $R^4$ backscatter/spreading corrections are applied, respectively. The best-fit lines are used to calculate the attenuation $\eta$ (dB/m). $\varepsilon = 6.5$ is used for time-to-depth conversion.

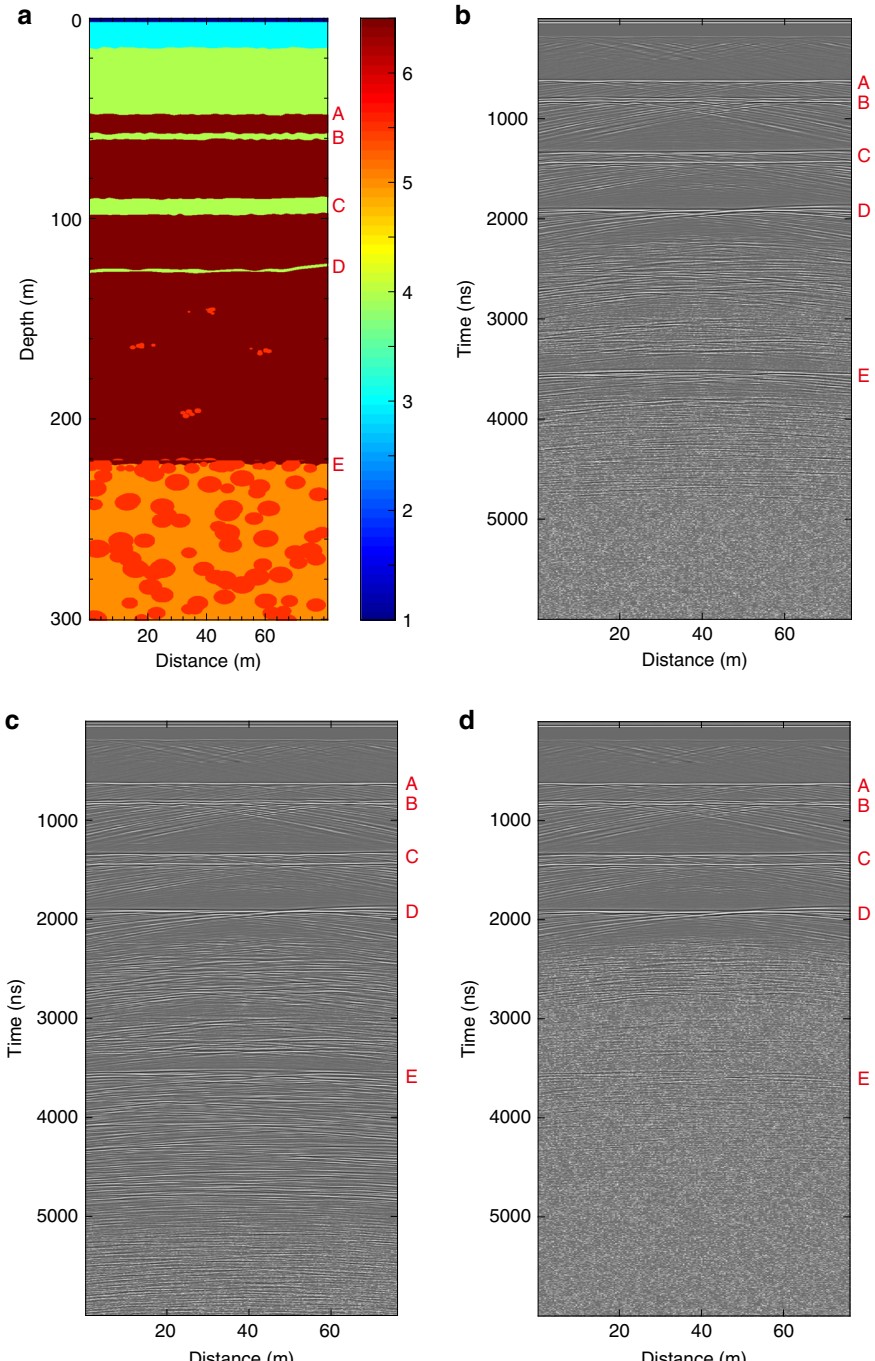

**Fig. 3 Radar simulation results. a** An 80 m-long path simulation model. **b** Simulation LPR result with loss tangent tanδ = 0.005. **c** Loss tangent tanδ = 0.001. **d** Loss tangent tanδ = 0.009. The color bar in **a** shows the permittivity value used for each layer. From top to bottom, the simulation model contains a vacuum, regolith layer, ejecta, and multiple basalt layers with interlayered regolith. In **a**, both small and large amount of ejecta is modeled at depth ranges of 120–220 m and 220–300 m, respectively.

characteristics to the CE-4 LPR radargram, confirming the plausibility of the subsurface model. The other two models (Fig. 3c, d) illustrate how different loss tangent values affect the radargram: Fig. 3c shows that the penetrating depth of the case of tanδ = 0.001 is 6400 ns, much deeper than 4900 ns in the case of tanδ = 0.005, and reflections from the ejecta are clearly visible, while LPR attenuates faster with tanδ = 0.009 (Fig. 3d), the reflected signals become weak beneath 3000 ns. Simulation results with different permittivity values of mare basalt are given in the Supplementary Note 4 and Supplementary Fig. 5.

**Trend surface analysis.** The CE-4 exploration path crosses old surfaces scattered with different sized craters, as revealed in the surface Digital Elevation Model (Fig. 4). The subsurface echoes marked as A–E are derived from aggregating all the repeated data tracks collected at the same waypoint. Their depth profile reflects lateral variations of the subsurface structure along the traveling path. The topographical variation within each subsurface layer is no larger than 8 m, averaging about 4 m. The trend surface analysis method is used to estimate the relatively large-scale systematic variation of the subsurface layer (Fig. 4). The arrows

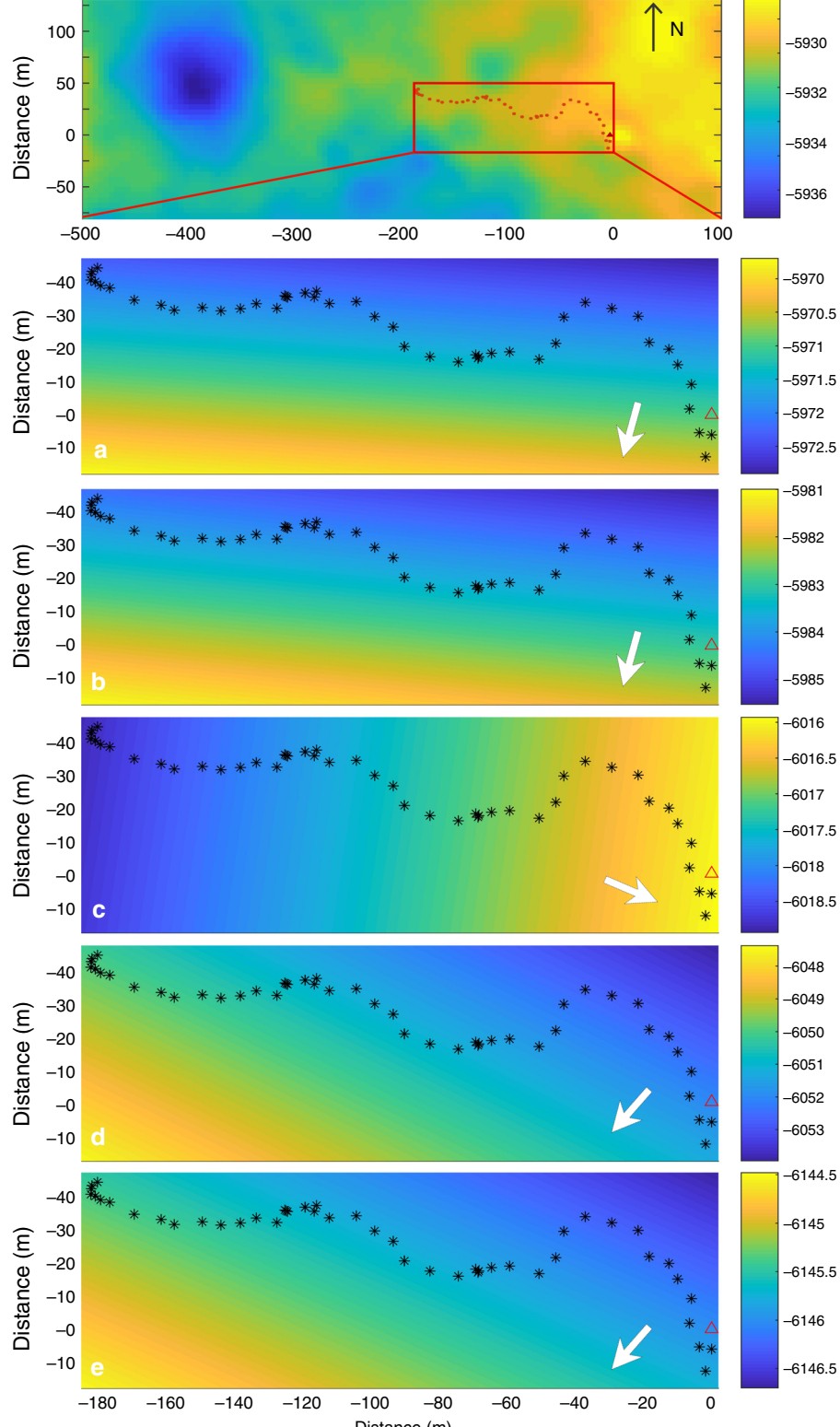

**Fig. 4 Trend surface analysis results.** Top image shows the surface DEM. A–E are the trend surface analysis of the subsurface structure of the CE-4 surveying area, corresponding to Fig. 1. The color bars indicate the elevation variation within each layer and the arrows show the rising direction of the layer. The coordinate system is based on the lander location marked by the red triangle. The black dots represent the rover's path during the initial nine months. Surface DEM data were derived from Lunar Reconnaissance Orbiter (LRO) Narrow Angle Camera (NAC) stereo pair image[28]. The positive direction of $Y$ axis points to the north and the positive direction of $X$ axis points to the East.

**Table 1 Estimations of the thickness of ejecta deposits.**

| Source | Ejecta (m) | Local material (m) | Ejecta deposits (m) | Great circle distance (km) | Age (Ga) |
|---|---|---|---|---|---|
| Pre-Nectarian epoch | | | | | |
| Hess | 1.2 | 1.5 | 2.7 | 281 | |
| Nectarian epoch | | | | | |
| Ingenii | 98.2 | 195.4 | 293.6 | 484 | 3.91[36] |
| Leibnitz | 63.2 | 63.7 | 126.9 | 222 | 3.88[36] |
| Davisson | 1.3 | 1.6 | 2.9 | 283 | |
| Boyle | 0.4 | 0.4 | 0.8 | 238 | |
| Abbe | 0.2 | 0.3 | 0.5 | 371 | |
| Imbrium epoch | | | | | |
| Imbrium Basin | 18.3 | 276.3 | 294.6 | 4996 | ~3.85 |
| Orientale Basin | 15.5 | 118.2 | 133.7 | 2269 | <3.85 |
| Maksutov | 0.5 | 0.8 | 1.3 | 336 | |
| Alder | 8.0 | 5.2 | 13.2 | 134 | |
| Eratosthenian and Copernican epochs | | | | | |
| Finsen | 4.7 | 3.2 | 7.9 | 140 | |
| Von Kàrmàn L | 1.5 | 0.6 | 2.1 | 72 | |

The ejecta deposits are the mixture of ejecta delivered by impact events and excavated local material. Great circle distance means the shortest distance on the lunar surface between the center of the crater and CE-4 landing site. Derivation formulas are described in the "Methods" section.

show that A, B, D layer rise toward the same direction, indicating possible sources for the lava flows located west of the lander, whereas layer C tends to rise toward the east, implying the potential source of an ejecta layer. E stratum also shows a prevailing rise at the western end, probably because it tends to become thicker in a westerly direction.

## Discussions

The data from the LPR CH-2 reveal that the first 38 m-thick section could consist of three distinct layers of fine-grained regolith, coarse ejecta, and fractured basalt. The bottom layer probably also features low-calcium pyroxene-bearing ejecta materials from neighboring craters, e.g., Finsen crater (Table 1), and autochthonous high-calcium pyroxene-bearing materials[19].

The first recognized reflector A in LPR CH-1 radargram (Fig. 1a) is found at the depth of 52 m; a uniform basalt layer may be present between 38 and 52 m. Alternatively, the possible boundary within this range is narrower than the CH-1 resolution, thus preventing detection.

Reflectors B, C, D probably represent interfaces between basalt layers from different periods, caused by the high permittivity contrast between solid mare basalt and high-porosity deposits formed during latent periods of lava activity. In this scenario, a stratum of regolith would have sufficient time to develop through a process of surface weathering, admixed with random ejecta deposits[12]. The trend surface analysis suggests that layer C rises towards the East, unlike layers A, B, D. This could be due to the presence of ejecta deposits delivered from the East, plausibly originating from crater Alder.

The 100 m-thick D–E stratum is interpreted as a layer formed by an undefined number of intermittent lava flows and small-scale ejecta deposits, possibly interposed by shallow regolith layers. This interpretation agrees with a prior study using small craters close to the CE-4 landing site, which found evidences of mare basalts at a depth between ~30 and 90 m[32].

The occurrence of large-scale, multiple lava flooding events within the VK crater is also revealed by several geomorphological features relating to a prominent dome located west of the crater. The elevation cross-sections of the 40 km mound/dome structure (Fig. 5b, c) suggest that it represents the last, less voluminous, and possibly more viscous lava flows that accumulated relatively close

to the vent/s, probably located at the foot of the crater terrace. Three finger-shaped flow lobes have heights of around 110 m (section a–b) with a slope of about 4.6°, whereas to the south and close to the rim, the flow also drops relatively abruptly by 100 m (section 1) with a 2.6° slope, which is comparable to the thickness estimation of the D–E stratum. These are substantial even in comparison with the well-studied mare Imbrium lobes, which range between 40 and 65 m[33,34]. This suggests that the infill history of the basin was punctuated and probably prolonged in time. However, much of VK's crater floor has been considerably scarred by countless secondary impact events since its formation, some very recently judging by the secondaries' size and freshness, rendering any attempt to estimate ages based on craters size-frequency distribution surveys arduous at best. However, given that these types of sharp geological boundaries have a long but limited lifespan on the lunar surface, it offers the intriguing possibility that the erupting activity might have continued into the Eratosthenian epoch, comparable to the Mare Imbrium 'young' flows[35]. Pasckert et al.[21] based on Neukum fits of the size-frequency distribution of craters within the VK crater floor derived a temporal interval between 3.75 and 3.15 Ga as the last eruptive phase. Using the same technique, extrusive events within the SPA basin are estimated to have peaked in the Late Imbrian period, ~3.74–3.71 Ga[36] ending about $3.6^{+0.09}_{-0.2}$ Ga[19].

The ejecta of the nearby fresh crater Zhinyu with a diameter of 3.8 km represent target materials down to about 300 m and the radial variation in olivine estimation content suggests the existence of at least three distinct layers with different olivine abundance (Fig. 6). The excavation process from impacts produces an inverse stratigraphy of ejected materials, with those closest to the crater rim representing the deepest part of the excavation. The spectrally derived olivine %wt abundances[37] reveal at least three types of potentially heterogeneous composition as shown by concentric circles marked in Fig. 6. Spectrally derived compositional data of shocked materials should be interpreted with a caution of course; however, the heterogeneous concentric pattern associated with Zhinyu is uncommon, thus making it more likely of reflecting actual compositional/petrological differences with depth.

The deepest strata seen by the LPR represent the large-scale ejecta deposits located at the depth of about 230 m with thickness more than 116 m, which is within the range of estimated ejecta

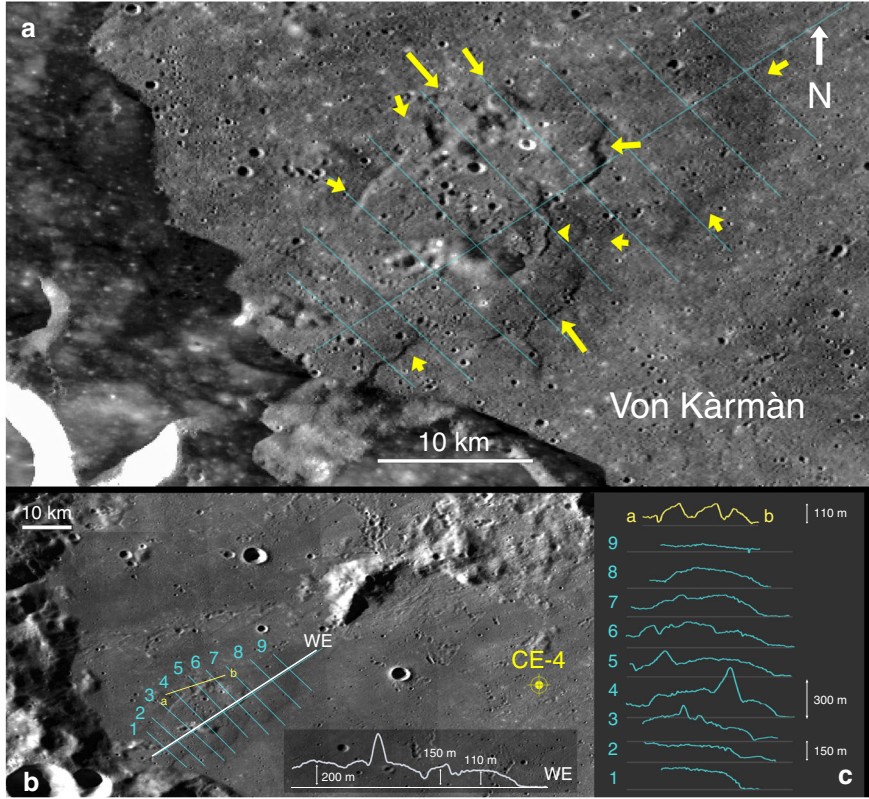

**Fig. 5 Western Von Kàrmàn flow structure.** Lobate fronts and other morphologic details of the flow structure are shown in the figure. Cross-sections altimetry data were generated from SLDEM2015 + LOLA data maps[50]. Centre lat-lon coordinates for image **a** are: 45.45°S,174.10°E, respectively. Arrows point at prominent flow fronts. LROC Wide Angle Camera (WAC)[51] image mosaic.

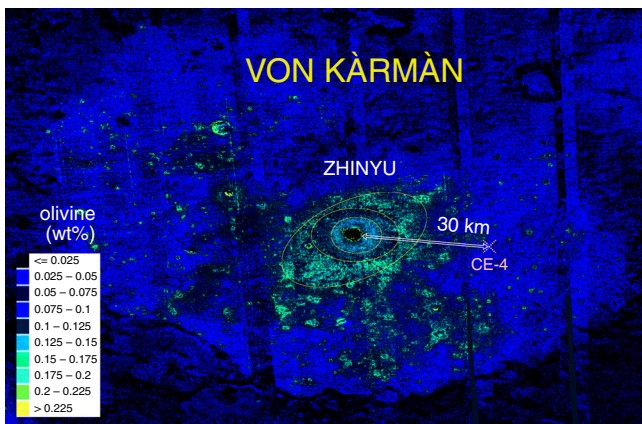

**Fig. 6 Olivine abundance (wt%) map mosaic of the VK crater.** Crater Zhinyu (3.8 km) displays an extended ejecta apron characterized by three different spectral signatures interpreted as relating to different olivine content of the radially distributed materials[37].

deposits from several source candidates (Table 1). The subsurface structure indicates the ejecta might come from the western direction, as it becomes thicker toward the right end of the radargram series (Fig. 1). Based on the analysis of the timing sequence of impacts, estimated thickness values, and the direction of the ejecta source, one possibility is that the strata E were emplaced by the Imbrium impact event or a mix of both Imbrium and Orientale. Based on broader geological considerations[18], the fact that the CE-4 landing site coincides to the antipodal position of the Imbrium impact suggests that the geologic unit Ig (Supplementary Fig. 1d) might be the product of its ejecta surge.

However, VK was heavily impacted by many later events and ejecta distribution is notoriously difficult to associate to a distal impact unless it is deposited within a clear ballistic path. Therefore, we cannot exclude other possibilities as sources of large-scale ejecta.

Based on the surface age derivations from size-frequency distribution of impact craters in the SPA basin, the volcanic activity appears to have peaked in the Late Imbrian. However, the start time of the volcanism in the region is not well constrained. The eruptive activities may have started as early as several million years later after the VK impact event (3.97 Ga). Later, large impact events such as Ingenii (~3.91 Ga), Leibnitz (~3.88 Ga), Imbrium, and Orientale, etc. together produced up to over 200 m of ejecta at the CE-4 landing site region (Table 1).

The mare infill of the basin probably followed this main deposition phase. This stage was punctuated by the arrival of small-scale ejecta from other distant impact craters or nearby relatively small craters. For example, the ejecta from Alder crater (Table 1) in the east might be buried by mare basalts at the depth of 96 m. In this scenario, the thickness of mare basalt would lie beyond the LPR CH-1 detection limit.

Using the excavation depths of the largest impacts that did not penetrate to the crater basement, the maximum thickness of the mare infill has been estimated to about 200 m[21] and possibly over 300 m. A higher estimate of 310 m was derived looking at the spectral characteristics of crater Zhinyv[19] ejecta, which is 32 km west of the CE-4 landing site. Our findings based on LPR observations also align with an overall basalt layer thickness larger than ~300 m.

Overall, the LPR data lead to an interpretative model of the local stratigraphy, which is comparable to that inferred from reflectance spectra data of crater ejecta[31]. The main difference

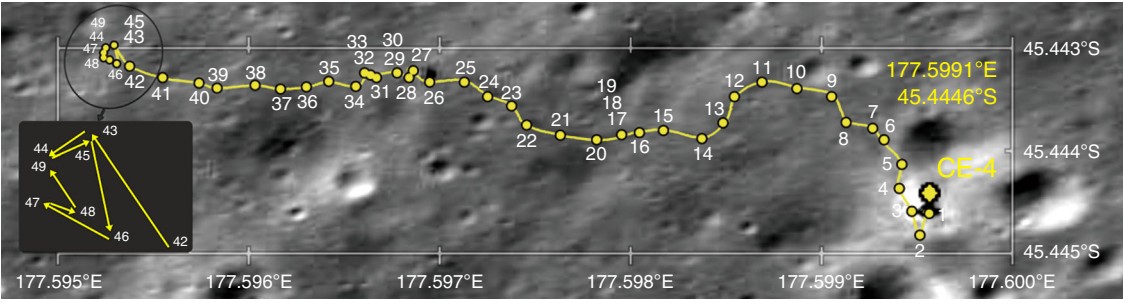

**Fig. 7 The routing path of the rover.** The yellow dots represent waypoints. The exploration phase from waypoint 42 to 49 is highlighted in the cartoon inset. The base image is Lunar Reconnaissance Orbiter Camera (LROC) Narrow Angle Camera (NAC) image M1311886645RC.

between the methodologies relates to the depth of the proposed layers, in the case of the LPR results, consistently deeper than previous estimations[19,21]. Another new insight is that the volcanism within VK was punctuated and prolonged, with at least four major infill events that can be interpreted from both the radargrams and geological considerations. The radargram provides direct evidence of multiple lava-infilling events having occurred within the VK crater, resulting in 12, 33, 34, and 96 m-thick lava layers at the CE-4 site. The radargram also shows that large-scale, multiple lava flooding occurrences were punctuated by the arrival of ejecta from impacts of different sizes and origin. In this work, we also derive an estimate for the average loss tangent of mare basalts on farside is inferred as 0.0040-0.0061.

## Methods

**Lunar-penetrating radar data**. CH-1 of LPR operates at the center frequency of 60 MHz with a 40 MHz bandwidth. The monopole antennas (12 mm in diameter and 1150 mm in length) are located at the back of the rover standing about 60 cm above the ground. In this work, we analyze LPR CH-1 data (file name list is given in Supplementary Table 1) collected in the first 9 months by the Yutu-2 rover along its 284.6 m-long exploration journey (see exploration path in Fig. 7).

**Data processing**. The radargram from the LPR CH-1 data was derived after removing repetitive data and background noise, applying filtering and amplitude compensation. Further processing details can be found in Supplementary Note 5 and Supplementary Figs. 6 and 7. The actual depth of the subsurface reflector, $d$, is converted from the two-way traveling time $t$ at the reflector and the relative permittivity of lunar basalt, $\varepsilon$, using

$$d = \frac{ct}{2\sqrt{\varepsilon}} \qquad (1)$$

The permittivity of the Apollo regolith and basalt samples[17,38], $\varepsilon = 4.5 (\leq 52\,\mathrm{m})$ and $6.5 (>52\,\mathrm{m})$ asured at 60 MHz is adopted in this work.

To identify the subsurface reflectors, not only the radargrams showing LPR data collected in the motion state were used but also the data trace from each waypoint (total 49 waypoints in this work, shown as yellow dots in Fig. 7) was generated by aggregating all the acquired repetitive data (~400–1000 tracks) at the same location to further reduce random noise and increase the signal-to-noise ratio. Furthermore, the identified reflectors using CE-4 LPR CH-1 data were compared with those derived from CE-3 data (Supplementary Note 6 and Supplementary Figs. 8–10) to avoid signal artifacts caused by the inherent system noise[39].

The trend surface analysis was performed with the reflector location of each waypoint and low-order polynomial fitting[40].

**Radar signal simulation**. To evaluate our geological interpretations based on the LPR radargram, simulation of the proposed subsurface stratigraphy model was conducted in the transverse electric mode with a two-dimensional finite-difference time-domain method using gprMax[41] and Gaussian noise set as the average signal level below the LPR CH-1 detection limit was included in the simulation. The detailed model and permittivity value of each layer are shown in Fig. 3 and Supplementary Note 4.

**Ejecta deposition estimation**. Large volumes of ejecta were delivered to the CE-4 landing site by several impacts[42]. The ejecta thickness was estimated using

$$T = 0.068 R_t (r/R_{at})^{-3} \qquad (2)$$

where $r$ is the distance from crater center to the landing site with the consideration of the curvature of the Moon and $R_{at}$ is the radius of a transient

cavity at the preimpact surface in meters[43,44]. For the complex craters,

$$R_{at} = 0.4906 (2R)^{0.85}, \quad R > 9.5\,\mathrm{km} \qquad (3)$$

$R$ is rim-to-rim radius of a final crater[45]. For the Imbrium and Orientale basins, which were formed after the VK crater, $R_{at}$ were obtained from Miljković et al.[46]. The apparent radius of Ingenii crater is 114 km[47]. The thickness of ejecta delivered to landing site is listed in Table 1.

The cratering efficiency ($\mu$) is the ratio between the thickness of local material excavated by the impact of ejecta and the ejecta. $\mu = 0.0092 r_{gc}^{0.87}$ (4) is adopted from Petro and Pieters[48], where $r_{gc}$ is the great circle distance. The thickness of ejecta deposits including ejecta and local excavated materials can be obtained from $h = T \times (1 + \mu)$ (5).

## Data availability

CE-3 LPR data and CE-4 LPR data are available at Data Publishing and Information Service System of China's Lunar Exploration Program (http://moon.bao.ac.cn/). All the LPR data IDs are listed in Supplementary Table 1. Data for Figs. 1 and 2 are available at https://doi.org/10.5281/zenodo.3763355. Data sources of Figs. 4–6 are given in the captions. Additional data related to this paper are available from the corresponding author upon reasonable request.

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

## Acknowledgements

Scientific data of Chang'e missions are provided by the China National Space Administration (CNSA). We are grateful for the support from the team members of the Ground Application and Research System (GRAS), who contributed to data receiving and pre-processing. This study is supported by the Science and Technology Development Fund (FDCT) of Macau (Grants 0042/2018/A2, 0089/2018/A3, 005/2017/A1, and 0079/2019/A2), the Pre-research Project on Civil Aerospace Technologies of CNSA (D020101), the Science and technology project of Jiangxi education department (Grant GJJ180489), and the Scientific Research Starting Foundation for scholars from Jiangxi University of Science and Technology (Grant jxxjbs18017).

## Author contributions

J.L.L., Y.X., and R.B. designed the research and wrote the paper. L.X. and X.P.Z. helped with the geologic analysis. J.J.L., X.M., and M.G.X. performed the calculations. B.L. and K.C.D. generated the Digital Elevation Model (DEM) of the Yutu-2 surveying path and helped with the NAC data processing. B.Z. and S.X.S. designed the instrument. L.Y.X. helped data calibration and mapping.

## Competing interests

The authors declare no competing interests.
