## [Peer Review File · Nature Communications]

Reviewers' comments:

Reviewer #1 (Remarks to the Author):

The authors discuss and interpret all observations made by the low-frequency (60 MHz) channel of the LPR ground penetrating radar on board the Yutu 2 rover. The main results of the LPR experiment are presented in this paper, which thus aims at becoming one of the main references for the study of the Chang'e 4 landing site. In my opinion, the scope and the subject of the paper are appropriate for publication in Nature Communications. However, I am concerned that at least some of the subsurface features visible in LPR radargrams could in fact be artifacts, as it happened on a twin experiment on board the Yutu rover of Chang'e 3. A quantitative analysis of LPR/Yutu data (Li et al., 2018) has shown that it is possible to distinguish between artifacts and real subsurface echoes, and I thus suggest the authors to consider performing such analysis also on their data.

* Main comment

In analyzing data from the LPR experiment on the Yutu rover, Li et al. (2018) note that "... artifacts and clutter are well visible on LPR data, especially on those collected with the 60-MHz dipolar antennas which were mounted above ground on the back of Yutu rover. In fact, the top part of the radar sections collected at this frequency on the Moon is systematically affected by a large ringing due to the antennas-rover coupling, whereas in the lower part, the radar signal is quite weak and difficult to be interpreted."

To my knowledge, the LPR experiment aboard Yutu 2 is a twin of the one on Yutu, with an identical mounting of the antenna and thus similar potential artifacts. The authors state in fact that "the top section of the radar signals becomes saturated due to the strong coupling effects from the electromagnetic interaction with the metal in the rover", which leads me to think that the analysis and caveats presented in Li et al. (2018) may well apply to the present paper. Quoting again from Li et al. (2018), "... such analysis demonstrates that the deep radar features previously ascribed to the lunar shallow stratigraphy are not real reflectors, rather they are signal artifacts probably generated by the system and its electromagnetic interaction with the metallic rover."

In particular, they find that radar horizons located at a time delay of 1100 ns, 2500 ns, 3700 ns, and 5800 ns from the surface are not real. Unfortunately, in the present paper (Figure 2) layer E appears to be located at about 3700 ns below the surface. For this reason, I would recommend a quantitative analysis based on the use of the S-transform, as in Li et al. (2018), or on any other similar methodology allowing the study of the spectral content of the signal.

* Other comments

In presenting simulations used to validate data interpretation, the authors report values for the dielectric permittivity but not for the loss tangent. Because this physical parameters differs significantly in basalts and in loose regolith, it seems to me that it would be interesting to discuss its effect on simulations and on their similarity to actual data.

The comparison between simulations and data presented in the paper is qualitative, and even if it is probably impossible to obtain a perfect match between them, because of necessary simplifications in the electromagnetic propagation modelling on which simulations are based, some discussion of the effects of changing values of the main simulation parameters (dielectric permittivity and loss tangent of individual layers) could in my view strengthen the authors' case for their interpretation of the radar signal.

In figure 4, the topographic rise visible in the lower left corner of panels showing the topography of layers A and B should be considered with extreme caution, in my view. This rise does not seem to contain data points, and it thus appears to be entirely produced by the interpolation algorithm used to build a regular topographic grid. Because different interpolation methods produce different results, this topographic rise may well disappear if a different interpolation method is used.

Reference 35, which is quoted in the text to predict that mare basalts should be at a depth of 30-90 m at the Chang'e-4 landing site, refers in fact to Chang'e-3. A recent estimate of the total regolith thickness at the Chang'e-4 landing site can be found in Di et al. (2019).

In Text S2, the authors state that they estimated the depth of penetration of the radar signal following the method presented in Xing et al. (2017). However, such method also provides estimates of the loss tangent. The authors themselves presented such estimates in a previous paper based on the high-frequency channel of LPR (Lai et al., 2019), and used them to distinguish between loose regolith and compact rock layers in the subsurface. Could the authors present and discuss their estimates of the loss tangent for low-frequency data?

Figure S1: if I interpret correctly, this figure shows raw radar data before processing. If so, there seems to be ringing noise not only at the top of the radargram, as discussed by Li et al. (2018), but also at the bottom: could the authors discuss its origin and its effect on data interpretation?

Figure S2: if I interpret correctly, this figure shows processed radar data which have been used to produce Figure 2a. The fact that the figure has a very bland colormap and no colorbar, and that the Y-axis reports sample number rather than time, makes it difficult to compare it with Figure 2a.

Figure S3: please add colorbar.

* References

Di, K. et al. 2019. Topographic evolution of Von Kármán crater revealed by the lunar rover Yutu-2. *Geophysical Research Letters* 46, 12,764–12,770.

Lai, J. et al. 2019. Comparison of dielectric properties and structure of lunar regolith at Chang'e-3 and Chang'e-4 landing sites revealed by ground-penetrating radar. *Geophysical Research Letters* 46, 12,783–12,793.

Li, C. et al. 2018. Pitfalls in GPR Data Interpretation: False Reflectors Detected in Lunar Radar Cross Sections by Chang'e-3. IEEE Transactions on Geoscience and Remote Sensing 56, 1325-1335.

Roberto Orosei

Reviewer #2 (Remarks to the Author):

This is an interesting paper demonstrating the subsurface imaging abilities of ground penetrating radar. The data shown in Fig 2 are wonderful! However, the accompanying science falls flat. For example, in the geologic settings there is lots of discussion about surrounding craters and their ages but that really is not discussed in the discussion. Conversely, lots of volcanism is discussed in the discussion but had never been introduced before. Thus, I think the paper would greatly benefit from a reorganization. My second major issue is that the GPR mapping is really not 3D maybe pseudo-3D at best. Furthermore, the authors fit's in fig 4 show major extrapolations that I believe were used in their interpretation. Maybe there was a typo (or I missed something) but reflector C and D clearly decrease and increase in elevation, respectively, against the authors interpretations. Overall, I would suggest this manuscript to majorly reworked, so I believe I have to reject it as Nature has such high standard. However, this is not because the data is not intriguing as clearly, it's just a matter of when this work gets published.

-David Stillman

My detailed comments can be found in the uploaded word document.

Reviewers' comments:

Reviewer #1 (Remarks to the Author):

The authors discuss and interpret all observations made by the low-frequency (60 MHz) channel of the LPR ground penetrating radar on board the Yutu 2 rover. The main results of the LPR experiment are presented in this paper, which thus aims at becoming one of the main references for the study of the Chang'e 4 landing site. In my opinion, the scope and the subject of the paper are appropriate for publication in Nature Communications. However, I am concerned that at least some of the subsurface features visible in LPR radargrams could in fact be artifacts, as it happened on a twin experiment on board the Yutu rover of Chang'e 3. A quantitative analysis of LPR/Yutu data (Li et al., 2018) has shown that it is possible to distinguish between artifacts and real subsurface echoes, and I thus suggest the authors to consider performing such analysis also on their data.

We thank reviewer's valuable suggestions. We have added substantially quantitative analysis of Chang'e-4 and Chang'e-3 LPR channel one data as follows.

* Main comment

Q1: In analyzing data from the LPR experiment on the Yutu rover, Li et al. (2018) note that "... artifacts and clutter are well visible on LPR data, especially on those collected with the 60-MHz dipolar antennas which were mounted above ground on the back of Yutu rover. In fact, the top part of the radar sections collected at this frequency on the Moon is systematically affected by a large ringing due to the antennas-rover coupling, whereas in the lower part, the radar signal is quite weak and difficult to be interpreted."

To my knowledge, the LPR experiment aboard Yutu 2 is a twin of the one on Yutu, with an identical mounting of the antenna and thus similar potential artifacts. The authors state in fact that "the top section of the radar signals becomes saturated due to the strong coupling effects from the electromagnetic interaction with the metal in the rover", which leads me to think that the analysis and caveats presented in Li et al. (2018) may well apply to the present paper. Quoting again from Li et al. (2018), "... such analysis demonstrates that the deep radar features previously ascribed to the lunar shallow stratigraphy are not real reflectors, rather they are signal artifacts probably generated by the system and its electromagnetic interaction with the metallic rover."

In particular, they find that radar horizons located at a time delay of 1100 ns, 2500 ns, 3700 ns, and 5800 ns from the surface are not real. Unfortunately, in the present paper (Figure 2) layer E appears to be located at about 3700 ns below the surface. For this reason, I would recommend a quantitative analysis based on the use of the S-transform, as in Li et al. (2018), or on any other similar methodology allowing the study of the spectral content of the signal.

A: CE-3 and CE-4 did use the same LPR equipment except that the mounting heights of two LPRs are a little bit different as rover design changed for the CE-4 mission.

Detail discussion about radar horizons located at a time delay of 3700 ns in CE-4 are in Supporting Information Text S4.

Text S4. Subsurface interface reflection in CE-4

Li et al. (2017) found that the deep radar features of channel one (CH-1) of CE-3 LPR might represent signal artifacts; therefore, we performed a quantitative analysis from three aspects to evaluate the radar features of the CE-4 LPR identified in our work.

The locations of horizontal continuous radar features.

As mentioned by Li et al. (2017), weak continuous signals at various depths of 1100, 2500, 3700, and 5800 ns (Figure S5a) appear in the LPR results of the ground experiments performed on glacier and artificial lunar soil, which cast doubts on the 3700 and 5800 ns features visible in CE-3 LPR results.

Figure S5 Identified radar features from (a) LPR results on Earth Li et al., (2017) (b) CE-4 LPR results. Y axle indicates the depth of the recognized radar features in ns.

Figure S5b shows the CE-4 LPR results, the reflectors identified by us locate at depth of (A) 723 ns, (B) 935 ns, (C) 1575 ns, (D) 2097 ns, and (E) 3720 ns. Most do not occur at the depth as revealed by signal noises from the ground tests. Although the 3720 ns is close to the depth of 3700 ns, as one of the suspected system noises of CE-3 and ground test results, their characteristics differ: the ringing noise in the ground test (Fig. S5a) is quite straight and

continuous, while several short, less continuous and curved signals appear around the reflector in CE-4 (Fig. S5b). If the feature were caused by system noise, it usually continues to be present from the left to the right in the radar image; however, some linear features in CE-4 are quite short and do not cover the entire roving path.

Noise level

(a)

(b)

Figure S6 Aggregated data track of (a) CE-3 LPR data (b) CE-4 LPR data

The signal-to-noise ratios (SNRs) of CE-3 (Fig.S6a) and CE-4 (Fig.S6b) LPR CH-1 data are estimated around 3700 ns, here:

$$SNR = 10 * \log_{10}\left(\frac{Signal^2}{Noise^2}\right)$$

SNR (CE-3) < 2

SNR (CE-4) = 13.6

Li et al. (2017) mention that the signal amplitude level of the suspicious reflector around 3700 ns and 5900 ns of CE-3 results is close to the noise level, while the SNR of the identified reflector of CE-4 at 3720 ns is obviously much higher than CE-3 LPR results.

Time–Frequency Analysis Applying S-Transform

(b)

Figure S7 S-transform results of (a) CE-3 LPR CH-1 data Li et al., (2017) (b) CE-4 LPR CH-1 data

The time-frequency analysis was performed with S-transform method, adopted from Li et al., (2017). The CE-3 results show that the central frequency (10-15MHz) of 3600ns-4000ns section is well below the original designated frequency (60 MHz), which also happens in the two ground test cases. Whereas the central frequency of CE-4 in 3600 ns- 4000 ns section is 35-45 MHz, different from CE-3 and ground test results.

* Other comments

Q2: In presenting simulations used to validate data interpretation, the authors report values for the dielectric permittivity but not for the loss tangent. Because this physical parameters differs significantly in basalts and in loose regolith, it seems to me that it would be interesting to discuss its effect on simulations and on their similarity to actual data.

A: Both permittivity values and loss tangent values are added to the simulation section, please see Fig. 2 and supplementary material Text S5. Different sets of parameters are used to show their effect on simulation, please see more details in Q3.

Q3: The comparison between simulations and data presented in the paper is qualitative, and even if it is probably impossible to obtain a perfect match between them, because of necessary simplifications in the electromagnetic propagation modelling on which simulations are based, some discussion of the effects of changing values of the main simulation parameters (dielectric permittivity and loss tangent of individual layers) could in my view strengthen the authors' case for their interpretation of the radar signal.

A: We add the simulation results of different loss tangent values in section of simulation and appendix as follows.

Simulation results

The average loss tangent value of the mare basalt in CE-4 site inferred from the penetrating depth of LPR is 0.005. Another approach using the geometrical spreading correction in the case of rough interface also derives a loss tangent of 0.0051 ± 0.0001 (See calculation processes in Q6). The first model (Fig. 2a) adopts the derived loss tangent value and models the subsurface structure underneath a short path, including regolith, ejecta from nearby craters, basalt layers admixed with small-scale ejecta, and large-scale ejecta formed in a time sequence from recent to remote. ...

The other two models (Fig. 2c and Fig. 2d) illustrate how different loss tangent values affect the radargram: Fig. 2c shows that the penetrating depth of the case of $\tan \delta = 0.001$ is 6400 ns, much deeper than 4900 ns in the case of $\tan \delta = 0.005$ and reflections from the ejecta are clearly visible, while LPR attenuates faster with $\tan \delta = 0.009$ (Fig. 2d), the reflected signals become weak beneath 3000 ns.

Simulation results with different permittivity values of mare basalt are given in the supplementary material Text S5.

Text S5. The LPR simulation results applying different dielectric permittivity values of mare basalt

Figure S8 shows the simulation results applying different dielectric permittivity values of mare basalt (6.5 vs. 10) and loss tangent values (0.005 vs. 0.009). First, varied permittivity values result in the difference in apparent depth of the reflectors. For example, the reflector E in Fig. S8a ($\epsilon=6.5$) appears earlier than that in Fig. S8b ($\epsilon=10$) due to its faster propagation speed. However, we have little knowledge about the detailed stratigraphy of the landing site to determine the permittivity value. Other methods such as the crater geomorphology cannot differentiate the lava flows erupted in different periods to offer references. Another prominent feature we observe is that the reflected signal in the deep section is visible in Fig. S8b ($\epsilon=10$), but not in Fig. S8a ($\epsilon=6.5$) although the loss tangent values stays the same: this is because the large permittivity value increases the dielectric contrasts between mare basalt and paleoregolith or ejecta and the reflectance upon the interface. Therefore, the echo strength of Fig. S8b is higher than the one in Fig. S8a. The loss tangent values we obtained in the paper is under the assumption of $\epsilon=6.5$. Third, the hyperbola shaped signals in D-E section of Fig. S8a ($\epsilon=6.5$) become flat in Fig. S8b ($\epsilon=10$). Nonetheless, the shape is also affected by factors with high uncertainties, so we did not infer permittivity values using hyperbola method in this work.

Figure 2 (a) 80-m long path simulation model (b) simulation LPR result with loss tangent $\tan \delta = 0.005$. (c) loss tangent $\tan \delta = 0.001$. (d) loss tangent $\tan \delta = 0.009$. The color bar in (a) shows the permittivity value used for each layer. From top to bottom, the simulation model contains a vacuum, regolith layer, ejecta, and multiple basalt layers with interlayered regolith. In (a), both small and large amount of ejecta is modeled at depth ranges of 120-220 m and 220-300 m, respectively.

Figure S8 LPR simulation results of (a) $\epsilon(\text{mare basalt}) = 6.5$ and $\tan \delta = 0.005$, (b) $\epsilon(\text{mare basalt}) = 10$ and $\tan \delta = 0.005$, (c) $\epsilon(\text{mare basalt}) = 6.5$ and $\tan \delta = 0.009$, (d) $\epsilon(\text{mare basalt}) = 10$ and $\tan \delta = 0.009$.

Q4: In figure 4, the topographic rise visible in the lower left corner of panels showing the topography of layers A and B should be considered with extreme caution, in my view. This rise does not seem to contain data points, and it thus appears to be entirely produced by the interpolation algorithm used to build a regular topographic grid. Because different interpolation methods produce different results, this topographic rise may well disappear if a different interpolation method is used.

A: We have changed the method to represent our interpretation. Instead of interpolating and extrapolating data to obtain the topography of each subsurface layer, we use trend surface analysis method to estimate the large-scale systematic variation of the layer and provide a trend of the rising direction of each subsurface layer.

We have updated Figure 3 in the paper as shown below and use an arrow to indicate the rising trend of each layer. The corresponding analysis is added to section of “trend surface analysis” and “discussion”.

Figure 3 Top image shows the surface DEM. A-E are the trend surface analysis of the subsurface structure of the CE-4 surveying area, corresponding to Figure 1. The color bars indicate the elevation variation within each layer and the arrows show the rising direction of the layer. The coordinate system is based on the lander location marked by the red triangle. The black dots represent the rover's path during the initial nine months. Surface DEM data were derived from Lunar Reconnaissance Orbiter (LRO) Narrow Angle Camera (NAC) stereo pair image²⁶. The positive direction of Y-axis points to the north, and the positive direction of X-axis points to the East.

Q5: Reference 35, which is quoted in the text to predict that mare basalts should be at a depth of 30-90 m at the Chang'e-4 landing site, refers in fact to Chang'e-3. A recent estimate of the total regolith thickness at the Chang'e-4 landing site can be found in Di et al. (2019).

A: The reference [33] Qiao et al. (2016) is corrected to Qiao et al. (2019), which estimated that the depth of the basalt layer is at 30-90 m using the penetrating depth of neighboring craters.

[33] Qiao, L., Ling, Z., Fu, X. & Li, B. Geological characterization of the Chang'e-4 landing area on the lunar farside. *Icarus* 333, 37–51 (2019).

Q6: In Text S2, the authors state that they estimated the depth of penetration of the radar signal following the method presented in Xing et al. (2017). However, such method also provides estimates of the loss tangent. The authors themselves presented such estimates in a previous paper based on the high-frequency channel of LPR (Lai et al., 2019), and used them to distinguish between loose regolith and compact rock layers in the subsurface. Could the authors present and discuss their estimates of the loss tangent for low-frequency data?

A: We added the estimated loss tangent values in the section regarding simulation; the calculation process can be found in the appendix text S6 and Figure S10, and used the value for the simulation and compare the LPR data and simulation results show that large loss tangent causes that the penetrating depth becomes shallower.

Text S6. The estimates of the loss tangent

In Xing et al. (2017), they used the following radar equation to estimate the penetrating depth with different loss tangent values.

$$G = \frac{\lambda^2 \zeta_1 \zeta_2 \sigma}{(4\pi)^3 R^4} e^{-4aR}$$

Where, G is the system gain, ζ_1, ζ_2 are the penetrating attenuation between two medias along input and output direction, λ is the wavelength in the medium, σ is the backscatter cross section, e^{-4aR} means the attenuation in the medium, and a means the attenuation constant.

The attenuation constant a is calculated by:

$$\alpha = \frac{2\pi}{\lambda_0} \sqrt{\varepsilon} \left[\frac{1}{2} \left(\sqrt{1 + \tan^2 \delta} - 1 \right) \right]^{\frac{1}{2}}$$

$G = 152$ db (Li et al. (2017), Xing et al., (2017)), $f=60$ MHz, $\epsilon=6.6$. The loss tangent is no larger than 0.005 for a detection limit of 300 m. Based on our calculation, the penetrating depth of CE-4 LPR CH-1 is 330 m, meaning the loss tangent is less than 0.005.

Figure S10 Depth vs. Signal Power profile and after R2, R3, R4 backscatter/spreading correction. The best-fit lines are used to calculate the attenuation η (dB/m). $\epsilon=6.5$ is used for time-to-depth conversion.

We also estimate the loss tangent values with the method used in Lai et al., (2019), as shown in Fig. S10. For R^2 correction, $\tan \delta = 0.0060 \pm 0.0001$; for R^3 correction, $\tan \delta = 0.0051 \pm 0.0001$; For R^4 correction, $\tan \delta = 0.0041 \pm 0.0001$.

Overall, the loss tangent values obtained by both approaches lie within the range of the measured results from the Apollo samples and the LRS estimation results (Ono et al. 2009). The loss tangent obtained from the Xing et al., (2017) is closed to the results from the R^3 correction in the second approach, which considers the case of a rough and planar reflector to calculate the scattering sectional area. The reason is that both methods use the same approximation to model scattering sectional area.

Q7: Figure S1: if I interpret correctly, this figure shows raw radar data before processing. If so, there seems to be ringing noise not only at the top of the radargram, as discussed by Li et al. (2018), but also at the bottom: could the authors discuss its origin and its effect on data interpretation?

Figure S2 shows the raw LPR data. Some of horizontal linear features in Figure S2 are reflected signals LPR collected at the same location repetitively, not ringing noises. Therefore, we removed the repetitive data in the first step of the data processing.

The ringing noise does appear in the ground tests on glacier, Chang'e-3 and Chang'e-4 LPR experiments (Fig. 4 in(Zhang et al., 2014)), especially in the top section (<500 ns) where the

signals are saturated. The noise may come from the coupling effect from the electromagnetic interaction with the metal in the rover. The time–frequency analysis shows that spectrum central frequency of the noise is around 12 MHz, so most of them is removed by applying a band-pass filter and repetitive data processing (see Fig. R1). One downside is that the saturated data in the near-surface section make extracting the stratigraphy information in the top part difficult and unreliable. Therefore, we used the data of LPR high frequency channel for data interpretation of the top 35 m. The ringing noises also interfere with the real reflector so that we need to check other criteria such as the aggregated echo strength, the shape and continuity of the reflectors, etc. to ensure the reliability of the interpretation.

Figure R1 CE-4 LPR CH-1 radargram after filtering, repetitive data removal and amplitude compensation.

Q8: Figure S2: if I interpret correctly, this figure shows processed radar data which have been used to produce Figure 2a. The fact that the figure has a very bland colormap and no colorbar, and that the Y-axis reports sample number rather than time, makes it difficult to compare it with Figure 2a.

A: We added the color bar, change the Y-axis to time and X-axis to the indices of waypoints of Figure S3.

Figure S3. Assembled all the LPR motion data with 38 data files in Table S1.

Q9: Figure S3: please add colorbar.

We add color bar to Figure S4.

Figure S4. The estimated penetration depth of CE-4 LPR.

Reviewer #2 (Remarks to the Author):

This is an interesting paper demonstrating the subsurface imaging abilities of ground penetrating radar. The data shown in Fig 2 are wonderful! However, the accompanying science falls flat. For example, in the geologic settings there is lots of discussion about surrounding craters and their ages but that really is not discussed in the discussion.

Conversely, lots of volcanism is discussed in the discussion but had never been introduced before. Thus, I think the paper would greatly benefit from a reorganization. My second major issue is that the GPR mapping is really not 3D maybe pseudo-3D at best. Furthermore, the authors fit's in fig 4 show major extrapolations that I believe were used in their interpretation. Maybe there was a typo (or I missed something) but reflector C and D clearly decrease and increase in elevation, respectively, against the authors interpretations. Overall, I would suggest this manuscript to majorly reworked, so I believe I have to reject it as Nature has such high standard. However, this is not because the data is not intriguing as clearly, it's just a matter of when this work gets published.

We thank reviewer's valuable suggestions. The introduction and discussion on surrounding craters and volcanisms are reorganized. We add the introduction of volcanisms in "geological settings" and move original Fig. 1 and corresponding analysis of surrounding craters in supplementary material text S1 and Fig. S1. The pseudo-3D structure is changed to trend surface analysis of derived subsurface stratigraphy. The interpretation of the trend of layer C and D are also revised.

My detailed comments can be found in the uploaded word document.

Detailed modifications are listed below.

Introduction

Add resolution info of LPR CH-1. "1-2 m in the mare basalt of 60 MHz frequency"

Figure S1

Figure S1 (a) labels of craters are added to the image. The choice of profiles to be presented were carefully selected to highlight geological and temporal relationships. For instance, we see that Leibnitz and Davisson are 'level', suggesting a similar infill history. On the other hand, Alder is much deeper than all of the others. Also interesting is to stress how VKM stands some 800 m taller than its neighboring VK, something that cannot easily be inferred from other imagery. The profile data, as in this case, offers the reader additional tools to aid evaluation and further studies.

Figure S1 (b) The map is a topography map. Scale is added and descriptions are corrected.

Figure S1 (c) caption: Bouguer Gravity map (deg. 60 to 600)²⁹ -> Bouguer Gravity map (deg. 60 to 600)³⁰, including elevation profile for A-B

Figure S1 (d) The figure has been extensively reworked, including its caption (See below). In particular, the focus of interest (the KM region) has been restricted.

Also, the nomenclature has been made uniform between the old and more updated maps and labelling of some craters added to ease location (the two smaller maps above offer a full labelling, anyhow). The slight chromatic differences between maps are not due to transparency rendering but from the original. However, since the colors are within the same pantone type, such as 'light green' for EC and red for I, we decided to stick to the original variations.

Fig. S1d caption changes to: 2019 map of the landing site superimposed on the Wilhelms I-1047 geologic map³¹⁻³³; nomenclatures of geological units relate to the lunar chronology

system (Copernican 'C', Eratosthenian 'E', Imbrian 'I', Nectarian 'N', and pre-Nectarian 'pN', followed by description of materials (crater 'c' or mare 'm') and terminated by map's authors sub-classes classification (please refer to original maps for full descriptions and nomenclatures): All maps are draped over a Wide Angle Camera (WAC) global morphologic base map.

Results - Geologic Settings

Add volcanism discussion. "Von Kármán crater lies within the South Pole Aitken Basin (SPA), the second largest confirmed impact crater in the Solar System (about 2,500 km in diameter). The thermal history of the crater and its neighborhood thus should be interpreted within an atypical geological context¹⁹⁻²⁵. The region underwent a relatively prolonged phase of lava infill, which lasted about 200-600 Ma^{25,33}, with the youngest flows estimated between 3.15 and 3.7 Ga. This period in the history of the Solar System was characterized by a proposed enhanced bombardment phase (Late Heavy Bombardment, LHB³⁴), which produced several giant impacts including Imbrium on the nearside and Von Kármán's northern neighbor, crater Leibnitz (245 km in diameter). These impacts, and those that followed, produced ejecta materials that punctuated the infill and post-infill phase of the Von Kármán crater."

"However, new data show that the rim of VK in terms of elevation and mass (i.e., Bouguer gravity) might have reshaped its northern neighbor, thus swapping the age relationship." is deleted.

Results – LPR results

The penetrating depth of "LPR Channel one (CH-1)"

CH-2 is the high frequency channel of LPR, which has finer vertical resolution but much shallower penetrating depth than those of CH-1. Due to the signal saturation issue of CH-1, we use CH-2 data to reveal the subsurface structure in the 0-35m section and CH-1 data to show the 52-328 m section.

The text is modified to

However, channel two (CH-2) of the LPR data, "the center frequency of which is 500 MHz and can penetrate till ~35 m", can be employed to complement the profile of the close-to-surface section¹⁸

"(TYPE 1 in Fig.2b)" is removed.

Figure 2

We remove the red lines and type 1, type 2 marks in Fig. 2(b) because they are not closely related with the interpretation of radargram.

Caption

"This 'filtering' effect can be clearly seen in the D-E layer where pareidolia could easily lead to overinterpretation in the estimation of weak reflectors." Is removed.

“red lines are secondary peaks; celeste lines are subtle boundaries denoting differences in ‘stripe’ directions and sharpness” is changed to “celeste lines are subtle boundaries denoting differences in ‘stripe’ directions and sharpness. Dashed lines denote higher uncertainty in location. (c) The interpreted stratigraphy structure inferred from the LPR results.”.

Results- Simulation

We combine the two models into one and add simulation results with different loss tangent values.

The text is changed to:

To test our geological interpretation of the radar data, several subsurface models were designed for LPR simulation with various sets of loss tangent and permittivity values. The average loss tangent value of the mare basalt in CE-4 site inferred from the penetrating depth of LPR CH-1 is 0.005. Another approach using the geometric spreading correction in the case of rough interface also obtains that the loss tangent is 0.0051 ± 0.0001 (See supplementary text S6). The first model (Fig. 3a) adopts the derived loss tangent value and models the subsurface structure underneath a short path, including regolith, ejecta from nearby craters, basalt layers admixed with small-scale ejecta, and large-scale ejecta formed in a time sequence from recent to remote. The simulation results show the clear boundary between the regolith and mare basalts (Fig. 3b). Also, both small- and large-scale ejecta could produce clear radar echoes, especially when larger debris is present at depths of over 200 m, which would produce flatter horizontal lines instead of large hyperbola shaped signals. The echoes of small-scale ejecta appear shorter and less continuous, comparable to the observations of the LPR results in D-E section. The simulation results show similar characteristics to the CE-4 LPR radargram, confirming the plausibility of the subsurface model. The other two models (Fig. 3c and Fig. 3d) illustrate how different loss tangent values affect the radargram: Fig. 3c shows that the penetrating depth of the case of $\tan \delta = 0.001$ is 6400 ns, much deeper than 4900 ns in the case of $\tan \delta = 0.005$ and reflections from the ejecta are clearly visible, while LPR attenuates faster with $\tan \delta = 0.009$ (Fig. 3d), the reflected signals become weak beneath 3000 ns. Simulation results with different permittivity values of mare basalt are given in the supplementary material Fig. S10.

Figure 2 (a) 80-m long path simulation model (b) simulation LPR result with loss tangent $\tan \delta = 0.005$. (c) loss tangent $\tan \delta = 0.001$. (d) loss tangent $\tan \delta = 0.009$. The color bar

in (a) shows the permittivity value used for each layer. From top to bottom, the simulation model contains a vacuum, regolith layer, ejecta, and multiple basalt layers with interlayered regolith. In (a), both small and large amount of ejecta is modeled at depth ranges of 120-220 m and 220-300 m, respectively.

Results – Trend Surface Analysis

We rewrote the 3D geological structure section. Instead of extrapolating topography data of each subsurface layer, we used “Trend Surface Analysis” method to the large-scale systematic variation of the layer and provide a trend of the rising direction of each subsurface layer.

The text is changed to:

The CE-4 exploration path crosses old surfaces scattered with different sized craters, as revealed in the surface DEM (Fig. 3). The subsurface echoes marked as A-E are derived from aggregating all the repeated data tracks collected at the same waypoint. Their depth profile reflects lateral variations of the subsurface structure along the traveling path. The topographical variation within each subsurface layer is no larger than 10 m, averaging about 6 m. The trend surface analysis method is used to estimate the relatively large-scale systematic variation of the subsurface layer (Fig. 3). The arrows show that A, B, D layer rise toward the same direction, indicating possible sources for the lava flows located west of the lander, while layer C tends to rise toward the east, implying the potential source of an ejecta layer. E stratum also shows a prevailing rise at the western end, probably because it tends to become thicker in a westerly direction.

Discussions

Fig. 5: add a, b, c, direction and scale.

Fig. 6: add three concentric rings to denote the three different spectral signatures and scale. Caption is changed to “Olivine abundance (wt%) map mosaic⁴³ of the VK crater. Crater Zhinyu (3.8 km) displays an extended ejecta apron characterized by three different spectral signatures interpreted as relating to different olivine content of the radially distributed materials.”

Add explanation of different spectral signatures in text. “The spectrally derived olivine %wt abundances⁴³ reveal at least three types of potentially heterogeneous composition as shown by concentric circles marked in Fig. 5.”

Table 1 caption is changed to: “Table 1. Ejecta deposits estimations. Derivation formulas are described in the method section.”

Figure 3 Top image shows the surface DEM. A-E are the trend surface analysis of the subsurface structure of the CE-4 surveying area, corresponding to Figure 1. The color bars indicate the elevation variation within each layer and the arrows show the rising direction of the layer. The coordinate system is based on the lander location marked by the red triangle. The black dots represent the rover's path during the initial nine months. Surface DEM data were derived from Lunar Reconnaissance Orbiter (LRO) Narrow Angle Camera (NAC) stereo pair image²⁶. The positive direction of Y-axis points to the north, and the positive direction of X-axis points to the East.

Method

The permittivity values used in the paper.

The top 52 m include regolith, ejecta deposits and basalt. So we use an average permittivity of 4.5 here. We also add the reference [18] that estimate the near-surface permittivity value using LPR CH-2 data.

References

- Zhang, H.-B., Zheng, L., Su, Y., Fang, G.-Y., Zhou, B., Feng, J.-Q., et al. (2014). Performance evaluation of lunar penetrating radar onboard the rover of CE-3 probe based on results from ground experiments. *Research in Astronomy and Astrophysics*, 14(12), 1633–1641. <https://doi.org/10.1088/1674-4527/14/12/011>

Reviewer #1 (Remarks to the Author):

I have much appreciated the effort and the care that the authors took in addressing the concerns expressed in my review of the original manuscript, and I have found their replies convincing. My evaluation is that, aside from very minor editorial issues (for example, using "axle" instead of "axis" in the caption of Figure S5), the manuscript is mature for publication.

Roberto Orosei

Reviewer #2 (Remarks to the Author):

This version is much better than the previous and is a great example of how GPR can image to great depths! However, there are still many things that are unclear and need work. Additionally, while this is a great GPR paper, I'm not sure that it has grand appeal to the Nature reader. We know VK had volcanism from the surficial features, what does this paper show us that we didn't know before? That some lava flows are buried in the subsurface, but why is this important for the way we look at the Moon? Additionally, there is lots of good stuff in the supplemental that is not discussed in the main text, so a long format journal may be better.

My detailed comments are included in the edited copy of the manuscript.

General comments:

Fig 1b. It is not clear how parts of the end of the traverse are calculated. If one digs they can kind of figure out, but it need to be explained in much greater detail here.

The description of how loss tangent was found needs a significant rewrite.

Fig 3. I still dislike this figure. There is so much extrapolation here. I think to be more honest the authors should use a common scale of the x- and y-axis (i.e. so that circles appear as circles not ellipses).

Supplemental figures: The last time I published a paper I had to ensure that all supplemental figures were mentioned in the main article. Maybe it is different now, but I don't think so.

Data access: The majority of the data files (\geq to index 10) in Table S1 are not available. Maybe this is a typo or they are not available yet. I am not sure on the current policy of Nature, but most journals require that data are made available by time of publication (and maybe this will be)

Reviewers' comments:

Reviewer #1 (Remarks to the Author):

I have much appreciated the effort and the care that the authors took in addressing the concerns expressed in my review of the original manuscript, and I have found their replies convincing. My evaluation is that, aside from very minor editorial issues (for example, using "axle" instead of "axis" in the caption of Figure S5), the manuscript is mature for publication.

A: We thank the reviewer's suggestion. We have proofread the paper and supplementary material again and fixed the error.

Reviewer #2 (Remarks to the Author):

This version is much better than the previous and is a great example of how GPR can image to great depths! However, there are still many things that are unclear and need work. Additionally, while this is a great GPR paper, I'm not sure that it has grand appeal to the Nature reader. We know VK had volcanism from the surficial features, what does this paper show us that we didn't know before? That some lava flows are buried in the subsurface, but why is this important for the way we look at the Moon? Additionally, there is lots of good stuff in the supplemental that is not discussed in the main text, so a long format journal may be better.

A: We thank the reviewer's suggestions.

In a nutshell, this work represents the first report on the detailed structure of the subsurface in a key and unexplored planetary surface. The 300+ meters radar profile of a lunar mare surface with a resolution of 1-2 meters, certainly warrants a prominent mention in the geological literature. However, its significance is further boosted by the fact that the data are produced by the only, still working, mobile 'laboratory' on the lunar farside. Nature Communications would have the first bite at the apple of an expanding dataset, thus with a healthy citation potential. We make this remark not to appeal to any self-promoting or utilitarian use of science, but to stress the significance of our results.

The volume of basalt deposits reflects the time-integrated volcanisms and is an essential parameter of any thermal evolution model of the Moon. A better understanding of the volcanic history of the South Pole- Aitken (SPA) basin, within the context of the lunar farside, is fundamental to the understanding of the remarkable nearside-farside asymmetric distribution of mare basalts.

Remote sensing analysis can only derive the lower bound of the thickness of mare basalts with relatively high uncertainties by looking mostly at morphological and ejecta pattern attributes of impact craters.

Currently, no direct evidence of the volcanic history of VK crater indicates whether the mare deposits were formed by one episode of basaltic volcanism based on the uniform reflectance spectral characteristics or multiple lava-infilling events. The crater morphology approaches, and spectrum data could not infer the thickness timing variations of the mare deposits because they cannot differentiate the lava flows erupted in different periods.

LPR provides direct proof of the thickness and corresponding timing variations of several subsurface strata beneath the surveyed path, representing lava flows that probably occurred during the Imbrium Epoch. The radargram also shows the occurrence of large-scale, multiple

lava flooding events was punctuated by the arrival of ejecta from other distant impact craters or nearby relatively small craters, which helps to reveal the geologic history within VK and constrain the timing sequence of the impact events that delivered ejecta to VK.

Furthermore, the average loss tangent of mare basalts on farside is inferred in this work as 0.0040-0.0061. The estimation of loss tangent value is added to the main text and all the supplementary texts and figures are introduced in the main text.

We modify the conclusions. “Overall, the LPR data lead to an interpretative model of the local stratigraphy that is comparable to that inferred from reflectance spectra data of crater ejecta³¹. The main difference between the methodologies relates to the depth of the proposed layers, in the case of the LPR results, consistently deeper than previous estimations^{19,20}. Another new insight is that the volcanism within VK was punctuated and prolonged, with at least four major infill events that can be interpreted from both the radargrams and geological considerations. The radargram provides direct evidence of multiple lava infilling events having occurred within the Von Kármán crater, resulting in 12, 33, 34, and 96 m thick lava layers at the CE-4 site. The radargram also shows that large-scale, multiple lava flooding occurrences were punctuated by the arrival of ejecta from impacts of different sizes and origin. In this work, we also derive an estimate for the average loss tangent of mare basalts on farside is inferred as 0.0040-0.0061.”

The end of abstract is revised. “The radargram reveals several subsurface strata interfaces beneath the surveying path: buried ejecta is overlaid by at least four layers of distinct lava flows that probably occurred during the Imbrium Epoch, with thicknesses ranging from 12 m up to about 100 m, providing direct evidence of multiple lava-infilling events that occurred within the VK crater. From the radar data, we also derive an estimation of the average loss tangent of mare basalts as 0.0040-0.0061.”

General comments:

1. Fig 1b. It is not clear how parts of the end of the traverse are calculated. If one digs they can kind of figure out, but it need to be explained in much greater detail here.

A1: We add an explanation in Fig 1b. caption as follows: Please note that LPR data were not collected at a fixed speed. For example, the jump from 180-240 m at the end of X-axis is because LPR CH-1 collected much fewer data at the end of the traverse path than at the beginning stage when rover travelled around a small crater.

Two enlarged images to show the slope of end of two reflectors are added as supplementary Fig. S3.

2. The description of how loss tangent was found needs a significant rewrite.

A2: We add the description of loss tangent estimation and amplitude fitting as Figure 2 in the main article as follows:

The average loss tangent (the ratio of the imaginary and real part of permittivity, $\tan\delta$) in CE-4 site is inverted with three types of geometric spreading corrections, as shown in Fig. 2. For R^2 correction, $\tan \delta = 0.0060 \pm 0.0001$; for R^3 correction, $\tan \delta = 0.0051 \pm 0.0001$; For R^4 correction, $\tan \delta = 0.0041 \pm 0.0001$. In the case of rough interface (R^3 correction), we confirm the estimation value with the result inferred from the penetrating depth of LPR CH-1, which is 0.005 (See supplementary text S6).

Figure 2 Depth vs. Signal Power profile and after R^2 , R^3 , R^4 backscatter/spreading correction. The best-fit lines are used to calculate the attenuation η (dB/m). $\epsilon=6.5$ is used for time-to-depth conversion.

3. Fig 3. I still dislike this figure. There is so much extrapolation here. I think to be more honest the authors should use a common scale of the x- and y-axis (i.e. so that circles appear as circles, not ellipses).

A3: Fig.4 has been re-drawn with the same scale of X- and Y-axis. It only provides an estimation of the trend of the rising direction of each subsurface layer as indicated by the white arrow in the following figure. We did not claim the subsurface topography data in the paper.

Figure 4

4. Supplemental figures: The last time I published a paper I had to ensure that all supplemental figures were mentioned in the main article. Maybe it is different now, but I don't think so.

A4: We have checked every supplemental figure and text are mentioned in the main article. Fig. S1 is mentioned in the introduction, Fig. S2-4 are mentioned in LPR results, Fig. S5 is mentioned in Simulation results and Fig. S6-10 are mentioned in method.

5. Data access: The majority of the data files (\geq to index 10) in Table S1 are not available. Maybe this is a typo or they are not available yet. I am not sure on the current policy of Nature, but most journals require that data are made available by the time of publication (and maybe this will be)

A5: LPR data files will be gradually uploaded to the website. We upload the LPR data in Fig. 1 and 2 to Zenodo (10.5281/zenodo.3763355) and modified the data statement.

Detail comments:

D1. What is the farside estimate? If nothing has been listed then it makes this less of a strong argument. Also how big is a regional scale?

A6: Reference [4] did not provide a farside estimation. We delete the sentence.

D2. Hellas is 2,300 km and Utopia is 3,300 km are confirmed and bigger than SPA. Mars north polar basin and Procellarum are much bigger but not confirmed.

A7: Revised.

D3. Before you say the LF channel has a frequency of 60 MHz, but it is not clear that CH-1 is the LF channel. So I would either change for saying the LF channel previously or just give the frequency here.

A8: Channel one (CH-1) of LPR is mentioned in the beginning of the main article.

D4. X-axis in b just from 180-240, why? So I somewhat understand that this is because around waypoint 42 the rover conducted a local exploration mission. Did it not collect radar data? Did you somehow just remove all this data. You need to change something here. Why you ask? I was trying to compute the slope of D and the deepest light blue line but I have no idea what the X-axis is? Also, I believe you need to mention how you came up with the depth. I think this is from Fig 2, but I'm not positive.

A9: We did not remove any data. Fig. 1 shows all the motion data collected within the first nine lunar days.

LPR CH-1 collected much fewer data at the end of the traverse than at the beginning stage when the rover circled a small crater. We modified Fig. 1b caption and added two enlarged images to show the slopes of the mentioned reflectors. The description of depth conversion is added in Fig. 1 caption.

D5. I don't know where waypoint 42 is and reflector D is steeply sloping between 37 and 47. Maybe 42 is the second red vertical line...

A10: The two red vertical lines indicate waypoint 37 and 42, respectively. We add the explanation in the caption of Fig. 1(b).

D6. This seems out of order to me. Shouldn't you prove the loss tangent before claiming what it is. More to the end of the paragraph.

D7. This is not a very convincing argument. Don't you need to fit the amplitudes, like the yellow line in Fig 1a? Is Fig S10 not a much better argument for how you estimated loss tangent?

A11: We rewrote the description of estimation of loss tangent and moved Fig. S10 to the main text. The discussions of the influence of different loss tangent values on simulation results were requested by another reviewer, not to illustrate how to estimate loss tangent values. Please see details in above A2.

D8. Why are b, c, d in orange? Why not gray scale like Fig 1?

A12: The color of radargram in Fig. 3 and supplementary Fig. S5 has been changed to gray scale.

D9. X-axis should be labeled somewhere in the figure. Also, make the units the same in X and Y to make circles equal circles, not ellipses

D10. Its green on the top plot

A13: Fixed. Please see Fig. 4 in A3.

D11. Fig 5?

A14: No. Previous work used titanium content distribution map to estimate the depth of mare basalt, not the olivine abundance map given in Fig. 6.

REVIEWERS' COMMENTS:

Reviewer #2 (Remarks to the Author):

Minor revisions. I don't believe I need to see this again. See attachment for all minor comments and corrections.

We greatly appreciate reviewers' comments and suggestions.

Reviewers' comments:

1. Line 70, delete "it"

Fixed.

2. Line 72, add "the"

Fixed.

3. Line 105 and 109, change "about" to "~"

Fixed.

4. Line 121, I have no idea what a celeste line is and neither does google. I think you mean cyan line.

A: Change to "light blue".

5. Line 122, still really hard to see these lines (two red vertical).

A: We have modified Fig.1. The width of these two lines is increased.

6. Line 135, I do not see this C appears to be flat in the interpreted Fig 1b. D is perfect!

A: Remove reflector C in the text.

7. Line 158-159, Can you give the e' of the deeper reflector?

A: At deeper location, the strength of reflector becomes weaker so that noise level is close to that of radar signal and causes inaccuracy of estimation of signal attenuation. Therefore, we did not provide the loss tangent value at the deeper part.

8. Line 173, delete a space before "Also"

A: Fixed.

9. Line 187, A depth scale on b,c,d would be nice for the common reader to tie a to the radargrams, but this is not needed. Or you could label the reflectors like you did in Fig S5.

A: Add labels in Fig. 3.

10. Line 303, I believe you need a better caption here or better column labels. What is ejecta (m) and local material (m)? It appears from context that Ejecta Deposit (m) is the thickness of ejecta at the landing site. It might also be nice to have a distance from the crater to the landing site.

A: We add the distance information in Table 1. The descriptions of ejecta, local materials and ejecta deposits are added to the caption.

Comments in Supporting Information:

11. Line 95-96, Please define a low level. Is that 0.4 or -0.4?

A: Low level means that the absolute value is 0.5. The explanation is added to the Supplementary Note 2.

12. Line 98, No depth given here. Where this is referenced in the main article you claim 330 m. It would be nice to know where 330 m is on this plot. Maybe add: Colorscale is a correlation factor and is unitless.

A: We add the color scale description in the caption and the depth information to the y-axis of the figure.

13. Line 116, change “means” to “is”

Fixed.

14. Line 118-120, I think this should be attenuation rate not constant. Then given the 152 dB and 300 m one can find the rate that then gives the loss tangent. Please clear this up. The “a” should be alpha, according to the second equation but it is a in the first. Alpha is not defined. Why $\epsilon=6.6$ everywhere else you use 6.5?

A: we adopted the term “attenuation constant” from a reference (Xing et al., 2017). But we have changed it to attenuation rate. “a” is changed to alpha. $\epsilon=6.6$ is the parameter used in the reference, we used $\epsilon=6.5$ for our own calculation. The source references have been added next to “ $\epsilon=6.6$ ”.

15. Line 119, change “db” to “dB”

Fixed.

16. Line 120, change “large” to “larger”

Fixed.